# Evaluation of Conifer Wood Biochar as Growing Media Component for Citrus Nursery

**Filippo Ferlito [1],\*** , **Biagio Torrisi [1]**, **Maria Allegra [1]** , **Fiorella Stagno [2]**, **Paola Caruso [1]** and **Giancarlo Fascella [3],\***

1    CREA, Research Centre for Olive, Citrus and Tree Fruit, Corso Savoia 190, 95024 Acireale, Italy;
     biagiofrancesco.torrisi@crea.gov.it (B.T.); maria.allegra@crea.gov.it (M.A.); paola.caruso@crea.gov.it (P.C.)
2    CREA, Research Centre for Olive, Citrus and Tree Fruit, Via La Canapona 1 Bis, 47121 Forlì, Italy;
     fiorella.stagno@crea.gov.it
3    CREA, Research Centre for Plant Protection and Certification, S.S. 113–Km 245.500, 90011 Bagheria, Italy
\*    Correspondence: filippo.ferlito@crea.gov.it (F.F.); giancarlo.fascella@crea.gov.it (G.F.);
     Tel.: +39-095-765-3106 (F.F.); +39-091-909-090 (G.F.)

**Abstract:** (1) Background: The commercial sustainability of the citrus nursery industry involves cutting costs by using alternative planting substrates to replace (or partially replace) the conventional black peat. Conifer wood biochar was evaluated as a component of the growth medium in a commercial citrus nursery for Carrizo citrange seedlings. (2) Methods: Seven growth media mixtures (A–G) were tested. Each mixture consisted of 50% sandy volcanic soil with the remaining 50% made up as follows: A = black peat and perlite 1:1; B = biochar 1; C = black peat, perlite, and biochar 0.5:1:0.5; D = black peat and biochar 1:1; E = black peat, compost, and biochar 0.5:0.5:1; F = black peat, perlite, compost, and biochar 0.5:0.5:0.5:0.5; G = black peat and lapillus 1:1, this substrate, previously adopted by the hosting nursery, was the control. (3) Results: The best media for the rootstock studied here were those containing 25% biochar (mixtures D and E). In the deeper layers the substrate was more compact, and the roots were limited to the shallow layers of the pot. (4) Conclusions: Conifer wood biochar can be partly added in place of peat in growth media mixtures, thus reducing costs and ameliorating sustainability.

**Keywords:** black peat; Carrizo citrange; charcoal; nursery costs; sustainability

---

## 1. Introduction

Citrus tristeza virus (CTV) is a major pathogen of citriculture and is responsible for what are sometimes catastrophic economic losses wherever citrus is grown [1]. Citrus replanting is not well recorded, but it is estimated that over the next decade some 20,000 citrus hectares will need replanting, and the only chance of success is if sour orange rootstocks are replaced by pathogen tolerant ones [1]. Examples of the latter are the trifoliate orange (*Poncirus trifoliata* L.) and its hybrids. The citrus nursery industry must now implement strict CTV prevention strategies including routine analyses of plant material [2]. Obviously, this significantly raises nursery production costs. The commercial sustainability of tree production cycles also involves cutting costs by using alternative planting substrates to take the place of the conventional black peat [3], which is a limited resource becoming increasingly scarce and thus expensive. Among the alternatives to black peat is biochar, a solid material derived from the thermochemical combustion of biomass in the absence of oxygen [4].

Biochar application improves the structure and fertility of agricultural soils as a good amendment and is also an alternative product for use as a nursery substrate both for *Citrus* spp. and ornamental plants [5,6]. The biomass used in the biochar pyrolysis process is variable [7]. Many of the original

structures of the biomass persist in the biochar and these influence its structural features [8]. The physical and chemical characteristics as well as the yield of biochar, depend on the pyrolytic process [9] and, in particular, the pyrolysis temperature which greatly affects the main biochar properties [10]. Pyrolysis conditions, together with the feedstock used, seem to significantly influence the biochar particle size distribution [11]. The original cellular structure of the biomass materials contributes to the material's macroporosity [12]. Macropores are crucial in the soil environment, affecting root penetration and improving soil microbial condition [13]. These structural aspects of biochar are the starting points for the formation of mesopores and micropores in the soil [14]. Moreover, it affects topsoil depth and texture through changes in the adsorption surface, pore-size distribution, soil density, and biological properties [15]. The associated reductions in $N_2O$ emission from biochar-amended soils and reduced nutrient leaching allow reductions in the amount of fertilizer needed [16]. Furthermore, biochar improves soil particle aggregation, workability, swelling and dynamic shrinking, permeability, and cation exchange capacity [17]. Water storage capacity is also enhanced [18]. Several studies have recently been conducted on the effectiveness of biochar as an alternative to peat compared to as an alternative thermal treatment of organic wastes (hydrochar) [19]. Recent researches have focused on the benefits of biochar on seed development, plant growth, and crop yield [20,21]. Enhanced growing conditions are also associated with improved plant health and lower incidences of disease [22]. An additional benefit is the reduced bioavailability of heavy metals, important in some soils [23].

The research described here is based on the idea that the characteristics making biochar an excellent soil improver in field and forest environments may also be useful in nursery production systems when added as a component in growth media used in small-volume pots [24]. Although it is plausible that biochar will be useful in this situation, since growing conditions in pots in a nursery are extremely different from those in the field, it is necessary to confirm the usefulness of biochar to potting mixes under nursery conditions [25]. It is also necessary to determine biochar's usefulness with particular plants, species, and varieties. Hence, this trial assesses conifer wood biochar as a prospective option in growth medium, when combined with agricultural soil and other organic and inorganic materials in media for growing Carrizo citrange seedlings.

## 2. Materials and Methods

### 2.1. Site, Plant Material, Nursery, and Training System

Research was carried out in a commercial nursery that produces certified citrus trees. This nursery is situated in Mascali (Cinque Terre) on the east coast of Sicily, southern Italy which has a warm, dry Mediterranean climate. The geographical location is: 37°44' N, 15°11' E and 0 m above sea level.

Carrizo citrange (*C. sinensis* (L.) Osbeck x *P. trifoliata* (L.) Raf.) seedlings were grown in a screen-house covered with white shade-net (light transmission 30%). In March 2013 seeds were sown in black polyethylene (PE) seed trays filled with black peat. In mid-November 2013 seedlings were transplanted individually to round black PE container pots each with a volume of 6.5 L (25 cm height × 20 cm diameter). The plant density was 24 containers/m$^2$, with one seedling per pot. Irrigation was applied independently to each container. Otherwise all managements were in line with local practice. After replanting, the seedlings were fertilized every six months with a prolonged-release fertilizer containing nitrogen (N), phosphorus (P), potassium (K), and magnesium (Mg) in the ratio 16:8:10:2. The fertilizer also contained the main micronutrients. The Carrizo citrange seedling stocks were grafted in mid-June 2015 with the scion *C. sinensis* (L.) cultivar Tarocco Lempso. The time of the trial (c. 2 years) started when seedlings were transferred to pots on 20 November 2013 and finished four months after grafting on 30 October 2015.

### 2.2. Growing Media Composition and Experimental Design

Seven growth media (gm A–G) were prepared; each contained 50% sandy volcanic soil and the remaining 50% was made up as follows: A = black peat and perlite 1:1; B = biochar 1; C = black peat,

perlite, and biochar 0.5:1:0.5; D = black peat and biochar 1:1; E = black peat, compost, and biochar 0.5:0.5:1; F = black peat, perlite, compost, and biochar 0.5:0.5:0.5:0.5; G = black peat and lapillus 1:1. According to the United States Department of Agriculture (USDA) scheme [26], the used volcanic soil is classified as sand. Perlite (Perlite Italiana Srl, Corsico (Milan), Italy) and peat (Baltic peat, Varena, Lithuania) were purchased from commercial enterprises. A homemade mature compost from a mixture of residues of citrus industries and citrus-tree pruning wastes was used. Medium G, previously adopted by the hosting nursery, was the control. The biochar was obtained from conifer forest wood waste and produced by slow pyrolysis at 400 °C for 48 h. Lapillus is an inert porous material coming from volcanic rocks and obtained through industrial calcination at high temperatures. All growth medium components were thoroughly mixed before filling the containers. Treatments consisted of a randomized complete-block design with three replicates, where each replicate comprised 20 plants, for a total of 420 plants (7 media × 3 reps × 20 plants).

### 2.3. Physico-Chemical Characteristics of the Growing Media

Before potting, six containers per treatment were weighed. For each growing media the physico-chemical characteristics were investigated on three replicate samples at the time of transplanting (November 2013) and again at the last month of the trial (October 2015). The organic matter and total nitrogen contents were evaluated only before potting, but the pH and electric conductivity (EC) were recorded every 60 days.

The water content at field saturation was registered, then the water content at field capacity and at permanent wilting point were recorded using a pressure plate apparatus at suctions of 30 and 1500 kPa, respectively [27]. The total available water capacity was determined as the difference between the moisture retained at 10 and 1500 kPa of suction. The ring knife method described by Tian and colleagues [28] was used to measure bulk density, water-filled porosity, total porosity, and air space of the mixtures. Total nitrogen (g kg$^{-1}$) was measured by Kjeldahl digestion [29] and organic matter (OM) was measured by quantifying total organic carbon (TOC, mg kg$^{-1}$) according to Springer and Klee [30]. The pH and EC determinations were carried out according to previous works [31,32] using an HI 9813 portable EC meter (Hanna Instruments, Woonsocket, RI, USA) and an AB 15 pH meter (Thermo Fisher Scientific, Waltham, MA, USA), respectively.

### 2.4. Seedlings Growth Monitoring

Seedling growth was observed every 60 days from two weeks after transplanting (5 December 2013) until grafting (June 2015) and at the last month of the trial (October 2015). Seedling growth was determined by measuring the stem height and basal diameter of the plants. Destructive measurements were carried out on six seedlings per growing media to determine the dry matter partitioning between the main plant components: root system, taproot, shoot system, and leaves. Measurements were carried out two times: on 20 February 2015 (in winter) and on 20 April 2015 (in spring); these dates were 14 and 16 months after the seedling transfer into the pot, respectively.

### 2.5. Statistical Analyses

Data were analyzed using the application StatSoft 6.0 for analysis of variance (ANOVA) and mean separation by Tukey's honest significant difference (HSD) test. The Kolmogorov–Smirnov test was performed to evaluate the normality of dependent variables. Since the results were significant, and therefore the hypothesis violated, a transformation into square root was carried out to satisfy the assumption of normality.

## 3. Results and Discussions

### 3.1. Chemical and Physical Properties of Growth Media

The main chemical and physical properties of the growing media are listed in Tables 1–3. As can be seen, different parameters modify significantly in the trial. In Table 1, D, E, and F show a significant and higher value of N than A and C. However, all growing media showed similar values to those reported with composted pine bark [33]. The organic matter (OM) content was very low according to Abad and co-workers [34]. In particular, A and C had the lowest values (Table 1). For both growing media, results were proportional to the amount of perlite (25%), the inorganic component. For F and G (control) which included 12.5% and the 25% of perlite and lapillus, respectively, the OM decreased. Meanwhile, B, D, and E incorporated biochar without any inorganic materials, and these showed higher levels of OM. It is noteworthy that OM content was relatively constant when biochar was 25% and 50%. Biochar appears to supply OM quantities similar to black peat and compost. High OM can improve water availability and water capacity and reduce compaction.

**Table 1.** Growing media chemical properties observed at the start (November 2013) and end (October 2015) of the trial. Values are means (±standard deviation). The b letter in a column indicate significant differences (lowercase indicate $p < 0.05$, uppercase indicate $p < 0.001$) compared to the letter a and c based on Tukey's honest significant difference (HSD) test.

| Growing Medium [a] | Total Nitrogen (N) (g kg$^{-1}$) | Organic Matter (%) | pH | | Electrical Conductivity (dS/m$^{-1}$ 25° C) | |
|---|---|---|---|---|---|---|
| | Start | Start | Start | End | Start | End |
| A | 1.96 ± 0.07 b | 7.03 ± 0.03 c | 5.08 ± 0.22 b | 5.20 ± 0.22 b | 1.64 ± 0.02 C | 0.99 ± 0.04 A |
| B | 2.24 ± 0.05 ab | 16.96 ± 0.07 a | 7.00 ± 0.18 a | 5.30 ± 0.78 b | 1.52 ± 0.06 C | 0.41 ± 0.03 B |
| C | 1.68 ± 0.02 b | 8.69 ± 0.12 c | 6.00 ± 0.16 ab | 4.95 ± 0.20 b | 1.75 ± 0.04 C | 0.47 ± 0.10 B |
| D | 2.80 ± 0.04 a | 15.72 ± 0.18 a | 6.05 ± 0.14 ab | 5.02 ± 0.25 b | 1.67 ± 0.06 C | 0.69 ± 0.07 B |
| E | 2.94 ± 0.06 a | 16.34 ± 0.27 a | 7.36 ± 0.11 a | 7.37 ± 0.33 a | 3.15 ± 0.20 B | 0.85 ± 0.04 B |
| F | 3.08 ± 0.81 a | 11.07 ± 0.08 b | 7.56 ± 0.19 a | 7.18 ± 0.14 a | 4.37 ± 0.06 A | 0.85 ± 0.05 B |
| G | 2.24 ± 0.03 ab | 10.34 ± 0.03 b | 4.66 ± 0.17 b | 5.37 ± 0.11 b | 4.37 ± 0.08 A | 0.75 ± 0.06 B |

[a] Growing media was 50% sandy soil + 50% of the following: A = black peat and perlite 1:1; B = biochar 1; C = black peat, perlite, and biochar 0.5:1:0.5; D = black peat and biochar 1:1; E = black peat, compost, and biochar 0.5:0.5:1; F = black peat, perlite, compost, and biochar 0.5:0.5:0.5:0.5; G = black peat and lapillus 1:1 (control, mixture adopted by the hosting nursery).

**Table 2.** Container weight, bulk density, and hydrological properties of growing media recorded at the start (November 2013) and end (October 2015) of the trial. Values are means (±standard deviation). Values in a column indicated by different letters are significantly different (lowercase $p < 0.05$, uppercase $p < 0.001$) based on Tukey's HSD test.

| Growing Medium [a] | Container Weight (kg) | Bulk Density (g cm$^{-3}$) | | Water at Field Saturation (%) | | Water Content at Field Capacity (%) | | Water Content at Permanent Wilting Point (%) | | Total Available Water (%) | |
|---|---|---|---|---|---|---|---|---|---|---|---|
| | Start | Start | End | Start | End | Start | End | Start | End | Start | End |
| A | 6.24 ± 0.27 B | 0.73 ± 0.05 AB | 0.72 ± 0.10 a | 38.76 ± 1.49 B | 41.02 ± 4.30 | 17.71 ± 2.73 BC | 20.01 ± 0.16 BC | 15.94 ± 0.26 | 14.16 ± 1.57 | 3.76 ± 2.82 B | 5.85 ± 1.53 BC |
| B | 6.15 ± 0.31 B | 0.63 ± 0.11 AB | 0.60 ± 0.08 ab | 45.18 ± 2.74 AB | 45.53 ± 0.61 | 25.71 ± 1.24 A | 26.47 A ± 1.35 A | 15.55 ± 1.10 | 14.41 ± 1.36 | 10.16 ± 1.54 A | 12.05 ± 0.25 A |
| C | 6.32 ± 0.46 AB | 0.64 ± 0.14 AB | 0.64 ± 0.05 ab | 41.31 ± 1.43 AB | 44.45 ± 3.47 | 20.70 ± 1.54 ABC | 21.86 ± 0.55 B | 16.30 ± 0.88 | 13.62 ± 1.02 | 4.40 ± 1.83 AB | 8.23 ± 0.61 ABC |
| D | 6.13 ± 0.30 B | 0.56 ± 0.10 AB | 0.66 ± 0.03 ab | 47.78 ± 0.16 A | 43.78 ± 2.89 | 22.95 ± 0.65 AB | 23.31 ± 1.35 AB | 17.17 ± 0.53 | 14.05 ± 0.72 | 5.77 ± 0.25 AB | 9.25 ± 0.70 AB |
| E | 6.5 ± 0.19 AB | 0.84 ± 0.12 A | 0.58 ± 0.03 b | 40.50 ± 0.47 AB | 44.93 ± 2.00 | 20.82 ± 0.59 ABC | 22.47 ± 1.15 B | 14.92 ± 0.84 | 14.53 ± 0.89 | 5.90 ± 0.24 AB | 7.93 ± 2.02 ABC |
| F | 6.92 ± 0.13 A | 0.64 ± 0.10 AB | 0.69 ± 0.05 ab | 44.01 ± 4.08 AB | 41.52 ± 3.24 | 21.71 ± 1.74 ABC | 19.77 ± 0.92 BC | 15.95 ± 2.61 | 14.59 ± 0.65 | 5.76 ± 2.47 AB | 5.17 ± 0.61 BC |
| G | 5.96 ± 0.80 B | 0.46 ± 0.07 B | 0.57 ± 0.04 b | 45.72 ± 0.66 AB | 42.64 ± 4.65 | 16.33 ± 1.41 C | 17.52 ± 1.38 C | 14.29 ± 2.67 | 12.52 ± 1.46 | 2.05 ± 1.56 B | 4.99 ± 1.47 C |

[a] Growing media was 50% sandy soil + 50% of the following: A = black peat and perlite 1:1; B = biochar 1; C = black peat, perlite, and biochar 0.5:1:0.5; D = black peat and biochar 1:1; E = black peat, compost, and biochar 0.5:0.5:1; F = black peat, perlite, compost, and biochar 0.5:0.5:0.5:0.5; G = black peat and lapillus 1:1 (control, mixture adopted by the hosting nursery).

**Table 3.** Physical properties of growing media in the trial period. Values are means (±standard deviation). Values in a column indicated by different letters are significantly different (lowercase $p < 0.05$, uppercase $p < 0.001$) based on Tukey's HSD test.

| Growing Medium [a] | Total Porosity (vol %) | | Air Space (vol %) | | Water Filled Porosity (vol %) | |
|---|---|---|---|---|---|---|
| | Start | End | Start | End | Start | End |
| A | 28.65 ± 1.99 | 18.73 ± 1.59 | 21.03 ± 2.00 ab | 12.94 ± 1.36 A | 8.62 ± 1.21 ab | 5.80 ± 0.23 B |
| B | 28.22 ± 0.69 | 19.09 ± 1.31 | 18.18 ± 3.27 b | 11.01 ± 0.90 AB | 10.04 ± 2.59 a | 8.08 ± 0.46 AB |
| C | 27.560 ± 1.95 | 18.31 ± 1.86 | 20.13 ± 4.12 ab | 10.27 ± 0.59 AB | 8.43 ± 2.25 ab | 8.04 ± 1.46 AB |
| D | 24.375 ± 3.67 | 18.94 ± 0.79 | 17.31 ± 1.52 b | 8.94 ± 1.18 B | 10.06 ± 2.15 a | 10.00 ± 1.56 A |
| E | 24.51 ± 1.48 | 16.58 ± 1.97 | 16.78 ± 0.80 b | 8.82 ± 0.91 B | 8.73 ± 1.18 ab | 7.76 ± 1.87 AB |
| F | 24.28 ± 0.71 | 17.32 ± 0.70 | 17.06 ± 0.43 b | 9.53 ± 0.14 B | 8.23 ± 1.12 ab | 7.79 ± 0.82 AB |
| G | 27.11 ± 2.87 | 18.02 ± 0.80 | 23.43 ± 2.65 a | 12.96 ± 0.93 A | 7.37 ± 0.41 b | 5.06 ± 0.79 B |

[a] Growing media were 50% sandy soil + 50% of the following: A = black peat and perlite 1:1; B = biochar 1; C = black peat, perlite, and biochar 0.5:1:0.5; D = black peat and biochar 1:1; E = black peat, compost, and biochar 0.5:0.5:1; F = black peat, perlite, compost, and biochar 0.5:0.5:0.5:0.5; G = black peat and lapillus 1:1 (control, mixture adopted by the hosting nursery).

The carbon/nitrogen (C:N) ratio was always below 4.2 (B) (not reported). These values indicate improved N availability, resulting in a net release of N into the substrate solution and, consequently, offering the opportunity to reduce the fertilization rate. A too-high (C:N) ratio can result in N immobilization [35]. Before potting the different mixtures showed variable pH values according to their composition. The starting pH ranged from 4.66 (G) to 7.56 (F) (Table 1). The E and F compost mixture and that containing 50% biochar (B) had a significantly higher pH than A and G. Only C and D had a pH suitable for citrus and mainly the Carrizo citrange rootstock examined here [36]. At the end of the trial, except for E and F, growing media containing biochar showed decreases in pH equal to 18% (C), 17% (D), and 24% (B), where the amounts of biochar were 12.5%, 25%, and 50%, respectively. These values reach the acceptable/optimum pH range of 5.3–6.5 [34] for most woody plants [25]. The growing media G (control) and A maintained lower pH values throughout the trial. In particular, at the beginning, the control was strongly acidic [36]; this may have reduced nutrient uptake. It is well known that pH has a strong effect on plant growth through its impact on the activity of beneficial bacteria. The soluble salt content, related to the EC of a saturated paste [37], assesses the ability of the soil water to carry an electric current. Thus, soil water EC is a good indicator of the quantity of nutrient mineral ions available for uptake. Initially, EC in G (control) was highest, followed by F and E. The EC in the other growing media was significantly lower (Table 1). The high levels of both pH and EC in compost-based growing media are reported [38]. Although the pH results agree with several studies [39–41], they are far from optimal for an ideal growing medium and could damage the growth of many acidophil and salt-sensitive species. The initial EC levels did not influence seedling growth, in accordance with previous results reporting good growth of a number of woody species in the presence of initial EC values exceeding 0.8 dS $cm^{-1}$ [42,43]. Heavy rainfall (typical for a Mediterranean climate) at the end of the trial in autumn caused leaching of nutrients as denoted by the EC decrease. Except for A, which showed a significant increase in EC, a drastic reduction in EC was observed for all other growing media. In particular, B and C (biochar 50% and 12.5%) registered the lowest values with respect to optimum ≤0.5 dS $cm^{-1}$ [34], but these were still well above the 0.06–0.2 dS $cm^{-1}$ recommended for healthy, vigorous growth [44,45].

The physical characteristics of the various growing media also changed based on their composition and the time of testing. The time-based changes occurred as a consequence of root growth as well as swelling and shrinkage associated with the cycling water status, and with decomposition of the organic components. Physical characterization of mixtures is very important to evaluate the percentage of solid material, water, and air capacity. Unlike natural soil, the volumes under investigation were very small (6.5 L) and very homogeneous, therefore, our sampling protocol provided relatively accurate estimates. The container weight (Table 2) was higher for F than for A, B, D, and G. A high bulk density increases transport costs and is associated with reductions in both porosity and air capacity [46]. Compaction increases a soil's bulk density, while reducing pore volume and water-holding capacity. At the start of the trial bulk densities ranged from 0.46 (G, control) to 0.84 (E), whereas the remaining mixtures were similar to the higher value (Table 2). At the end of the trial E had the lowest bulk density, comparable to G (control). At the end, the bulk density in the mixtures containing biochar decreased from 5% (B) to 44% (E), probably due to root growth and downward movement of fine particles to greater depths increasing from 7% (F) to 15% (D). For growing media C bulk density did not change significantly over the trial period, whereas for G (control) it increased by 19%. The value of bulk density in an ideal growing media should be <0.40 g $cm^{-3}$, with a value range between 0.1 and 0.3 g $cm^{-3}$ considered acceptable for seedling propagation [34,47]. At the end of the trial the mixtures containing biochar had higher bulk densities than optimum (0.40 g $cm^{-3}$), from 0.66 (B) to 0.58 (F) (Table 2). The high bulk density values in this trial are a result of their 50% soil component. The hydrological parameters are very important for water availability and also suitability for plant growth [28,48]. The water content at saturation at the start was highest for D and lowest for A, whereas at the end no differences were found among the seven mixtures. At the start and at the end of the trial the field capacity was highest for B (50% biochar) (Table 2). This was significantly worse than for G (control). No differences in permanent

wilting point were recorded among the seven mixtures. For all mixtures and at both the start and end, available water capacities were very low. In our trial, especially at the start, the low values were due to the uncompacted nature of our mixtures. At the end of the trial micropore values increased and growing medium B had significantly higher values than A, F, and G.

There were no differences between mixtures in porosity (Table 3). Except for D, E, and F, air space volumes at the end of the trial were less than 10% which is considered unacceptable for a greenhouse substrate [49]; the other growing media had ideal air porosities [50]. Growing media B (50% biochar) had the lowest decline (38%) in porosity which aligned with the bulk density values. Water filled porosity was highest for B and D at the start and D at the end of the trial with respect to G at the start and to A and G at the end of the trial (Table 3).

### 3.2. Evolution of Plant Growth

The seedlings height was significantly greater for G (control) than for B (Figure 1A). This agrees with previous results which report maximum growth and stem diameter for citrus seedlings in media containing a lower amount of peat (peat + sand 1:4) [51]. At the end of the trial, the grafted plants all reached similar heights (Figure 1A). The values of stem diameter were significantly greater in C than in B, E, and F. Seedlings cultivated in all biochar growing media were similar to control G (Figure 1B). In the grafted plants, stem diameter was greater in growing media C than in F. Also, seedlings in growing media containing biochar were comparable to G (Figure 1B).

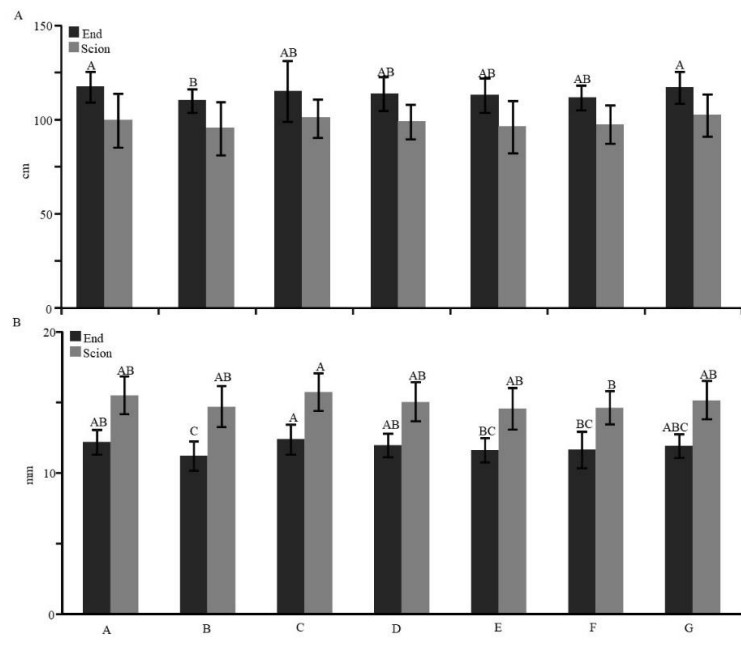

**Figure 1.** (**A**) Seedling height and (**B**) stem diameter at soil level measured on rootstocks in various growing media just before grafting and five months after grafting. For each parameter and period ANOVA and mean separation by Tukey's HSD test ($p \leq 0.01$; bars represent standard deviation). [a] Growing media was 50% sandy soil + 50% of the following: A = black peat and perlite 1:1; B = biochar 1; C = black peat, perlite, and biochar 0.5:1:0.5; D = black peat and biochar 1:1; E = black peat, compost, and biochar 0.5:0.5:1; F = black peat, perlite, compost, and biochar 0.5:0.5:0.5:0.5; G = black peat and lapillus 1:1 (control, mixture adopted by the hosting nursery).

The destructive analyses (Figure 2) during winter for all growing media show that the dry matter accumulated in leaves, stems, and taproots were similar for all mixtures. The differences did occur in root systems development. The highest dry matter accumulation was shown in A, with accumulation in B, C, D, and E being significantly lower. The root system in A and G growing media represented

20% of the total dry weight; whereas in the biochar growing media, except for F, the root systems accounted for 11% (B), 10% (C), 10.19% (D), and 13% (E) of total dry weight. These values do not completely agree with the findings of Girardi and colleagues [52] who found higher values of dry weights for one-year-old roots of Pera sweet orange tree grafted on Rangpur lime grown in commercial inert substrates. Therefore, the substrate and the slow-release fertilization did not affect the seedlings' growth rate. In spring, there were no significant differences between growing media for dry matter content of leaves and stems. However, the roots of A and G (control) differed significantly from D. The taproot dry matter in B was significantly lower than in A and G at the end of the trial. In spring, A, F, and G (control) showed a drastic reduction in root dry matter (13.62%, 12.91%, and 14.49%, respectively), while the other growing media showed a slight increase (Figure 2). In spring, the root/stem ratio ranged from 0.18 for D to 0.25 for G. The root dry mass reduction in spring was is in line with the findings of Bevington and Castle [53] who reported that in citrus trees, and in Carrizo citrange in particular, even in non-limiting soil and climatic conditions, roots' growth rate decreased during the period when shoot growth was particularly rapid. Therefore, root dry matter content reductions in spring in G (control), A, and F, were probably due to a greater degradation of secondary roots and hairs between winter and spring resulting in alternating patterns of root and shoot growth.

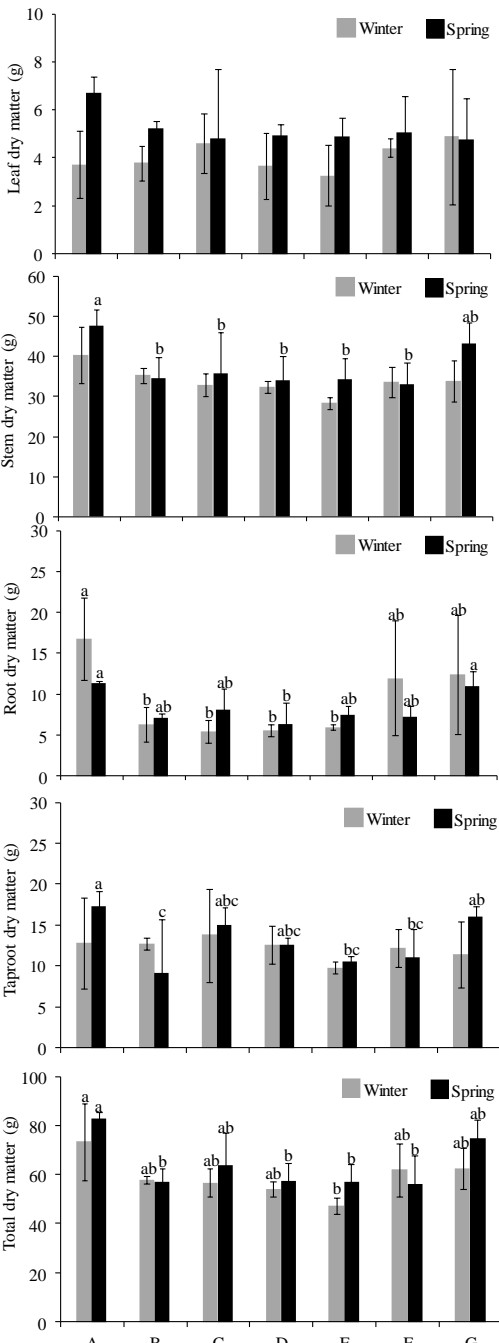

**Figure 2.** Dry matter distribution recorded in winter (February 2015) and spring (April 2015) of Carrizo citrange seedlings potted in different growing media. For each parameter and period ANOVA and mean separation by Tukey's HSD test ($p \leq 0.01$; bars represent standard deviation). The b and c letters above different bars indicate significant differences compared to the letter a. [a] Growing media was 50% sandy soil + 50% the following: A = black peat and perlite 1:1; B = biochar 1; C = black peat, perlite, and biochar 0.5:1:0.5; D = black peat and biochar 1:1; E = black peat, compost, and biochar 0.5:0.5:1; F = black peat, perlite, compost, and biochar 0.5:0.5:0.5:0.5; G= black peat and lapillus 1:1 (control, mixture adopted by the hosting nursery).

The lower root growth than aerial growth indicates an acceptable root system efficacy. However, the significant low root growth within the deepest layers of the pot could be caused by the reduction of substrate quality. As observed, this is due to the collection of small particles in the ground level of the pot thus reducing the porosity (not shown). With an increase in bulk density, the macroporosity is decreased,

which enhances the force necessary for the roots to deform and displace the substrate particles; at the same time root elongation rates decrease thus affecting root development and morphology [54]. In particular, when the substrate was more compressed, the roots placed themselves throughout the upper section of the pot so as to guarantee expansion of the hypogean apparatus, together with the related adequate root aeration. However, the root expansion reduction is not necessarily correlated with the low plant nutritional status [55].

In a ranking order, C and D (containing 25% biochar) presented the most effective growing media for citrus seedlings growth, showing similar or higher performances with respect to the benchmark control (G), as they provided optimal physical and chemical characteristics for plant development (Tables 1 and 2), resulting in higher root and shoot growth (Figures 1 and 2). In contrast, B, E, and F were less effective, showing lower root growth (Figure 2); in particular, the latter two growing media containing compost showed lower values of some growing parameters at the beginning of the experiment (Figure 1) and these negative responses were probably related to the higher salinity of the compost due in turn to its incomplete maturation.

## 4. Conclusions

Peat can be partially substituted by conifer wood biochar. The new growing media for citrus nurseries allows growth rates similar to those recorded for plants grown in conventional peat mixtures. Although 50% biochar did not affect seedling development, the proportion that seems optimal for the rootstock studied here seems to be 25% biochar. The coupling of biochar with the other matrices used was positive; although when mixed with compost, the performances of the seedlings worsened, due to the excessive improvement of the reaction of the substrate.

**Author Contributions:** Conceptualization, F.F. and G.F.; methodology, F.F., B.T., G.F., and M.A.; software, M.A., F.S., P.C., and F.F.; validation, F.F., M.A., P.C., and G.F.; formal analysis, F.F. and G.F.; investigation, F.F. and B.T.; resources, F.F. and G.F.; data curation, M.A. and F.F.; writing—original draft preparation, F.F. and G.F.; writing—review and editing, F.F., G.F., and F.S. All authors have read and agreed to the published version of the manuscript.

**Funding:** This research received no external funding.

**Conflicts of Interest:** The authors declare no conflicts of interest.

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
