# Peer review of "Evaluation of Conifer Wood Biochar as Growing Media Component for Citrus Nursery"

_applsci, doi:10.3390/app10051618_

Round 1
Reviewer 1 Report
It's critical that we begin to lessen our use of peat as its harvest is unsustainable. Biochar is a material that shows great promise as a replacement. This paper supports that application.
There are a few issues to fix before publication. I've attached a copy of the mss. with the first few pages reviewed for usage. Please be sure the rest of the mss. is likewise revised.
There are some unknown terms in the mss. What is "lapillus?" And in Figure 1, what are the bar labels?
On page 4, there is a statement that requires evidence (lines 156-157) or must be removed. On the same page, line 168, you state that there is a significant increase for EC on the first line. It's a decrease. Please resolve this issue.
In general, I was hoping for a more detailed discussion of the perceived benefits and disadvantages of each of the mixes, or at least a ranking of the mixes from best to worst. Currently all that's stated is that biochar can substitute for peat. It's a shame not to tease out other recommendations based on the tables and figures.

Reviewer 2 Report
The introduction is very brief. Authors should provide more background on studies with biochars and hydrochars in growing media. See The effect of sewage sludge biochar on peat-based growing media Hydrochars from biosolids and urban wastes as substitute materials for peat The effect of pruning waste and biochar addition on brown peat based growing media properties Line 87: Adequate soil taxonomy (FAO or USDA) should be provided. Sources of perlite, compost and peat should be provided, so that the experiment could be replicated. Line 92: Please add heating rate and duration. Lines 119-120: Did the data meet the assumptions for an ANOVA? Have you checked data normality and assumptions on variance? Lines 123-124: This sentence is not adding anything to the article. Please, delete. There is an excess of abbreviations. Please, do not abbreviate growing media as "gm" Line 279-280: I did not understand this sentence.Author Response
Please see the attachment

Reviewer 3 Report
Graphical results would be much better to make a comparison between the seven tests. Now it takes wading through a substantial amount of data.
